# Name use by companion parrots

**Lauryn Benedict**[1]*, **Viktoria Groiss**[2], **Marisa Hoeschele**[3], **Eva Reinisch**[3],
**Christine R. Dahlin**[4]

1 Department of Biological Sciences, University of Northern Colorado, Greeley Colorado, United States of America, 2 Department of Linguistics, University of Vienna, Vienna, Austria, 3 Acoustics Research Institute, Austrian Academy of Sciences, Vienna, Austria, 4 Department of Biology, University of Pittsburgh at Johnstown, Johnstown, Pennsylvania, United States of America

* lauryn.benedict@unco.edu

## Abstract

Humans organize social interactions in part by referring to others using proper names (hereafter "names"). Names might also facilitate the complex social lives of animals. Several animal species produce name-like signature sounds in nature and can vocally target interaction partners, but researchers hesitate to equate these sounds with the human linguistic concept of a name. A more direct way to ask if diverse species can learn names and use them appropriately is with animals that learn human words and phrases. Accordingly, we used survey data to determine whether parrots that live with humans regularly learn names and can potentially use them as individual vocal labels for people and animals. Survey takers were asked about word and phrase use by companion parrots; 47% of reports on 884 birds included examples of name use, with those 413 parrots speaking 802 phrases that included names. For a subset of these individuals, survey-takers provided contextual information that allowed us to assess whether parrots used names in ways consistent with vocal labeling. Parrots used names in a range of social situations, including greetings, separations, and when seeking attention. Reports on 88 different birds of 30 species suggested that parrots applied names appropriately as vocal labels for humans and animals, with strong evidence that some birds applied names only to single individuals and not as category labels. At the same time, many parrots used names in contexts outside of typical human linguistic conventions, such as seeking attention by vocalizing their own name. Results indicate that captive parrots learn and use names in a variety of situations, sometimes applying them as vocal labels when communicating with or about others. This suggests that parrots have the cognitive and vocal capacity to use names but leaves many open questions about how animals label individuals using vocal signals.

**Data availability statement:** All relevant data are within the manuscript and its Supporting Information files.

**Funding:** This work was funded in part by the Vienna Science and Technology Fund (WWTF) project ANIML (LS23-014) to MH. The funders had no role in study design, data collection and analysis, decision to publish, or preparation of the manuscript.

**Competing interests:** The authors have declared that no competing interests exist.

## Introduction

All human languages use proper names for individual people, places, and things [1]. In English (and many other languages), proper names are a type of label distinguished from other labels by their classification as proper nouns that each reference a single entity and are capitalized when written [2]. Proper names for people are learned vocal labels (sometimes called personal names) and are used by humans to organize social interactions both when speaking to someone and when speaking about someone, thereby recognizing the individuality of that person [1–3]. Human proper names are used similarly by multiple people within a social group, and they refer to an individual even when that person is not present [2]. Evidence suggests that proper names are learned differently from category labels, and require unique cognitive processing [4,5].

Biologists have looked for evidence of proper name use among a range of non-human species, debating whether or not animals use names for each other [6–10]. In line with previous animal behavior literature that generally uses the term "name" to represent proper names, we hereafter do the same. Although no researchers have found unequivocal name use among wild animals, they have demonstrated that animals have a range of strategies for acoustically identifying and addressing individuals in their social groups [6–8,10]. Diverse species can recognize individuals by the acoustic properties of their vocalizations (much as humans do when recognizing voices) [11–14]. Unlike name use, however, this ability allows a listener to recognize the vocalizing individual but does not allow a signaler to target a particular receiver. Egyptian fruit bat (*Rousettus aegyptiacus*) vocalizations contain information in their spectral structure that can identify both the sender *and* receiver of the call [15]. African elephant (*Loxodonta Africana*) vocalizations are sometimes used to address certain individuals, but researchers have been unable to isolate specific call features or standalone sound units that consistently identify individuals [8]. Marmosets modify the structure of "phee" calls according to their interaction partner in ways that allow for vocal labeling [10]. Parrots and dolphins have individual signature sounds that can be imitated by group-mates to refer to and address individuals [16–18]. Thus, all of these species have vocal elements that help animals target particular signal recipients, but researchers have generally not called these vocal signatures names [19]. This does not discount the possibility that animals can use names for individuals but to date we have not found names conclusively in wild populations, likely because we still have a very imperfect understanding of the information content conveyed via many animal signals [20].

One way to circumvent the challenges of understanding wild animal vocalizations is to test whether animals can readily learn, recognize, and use names as vocal labels within the framework of human language. Domestic animals of many species respond behaviorally to their own names, suggesting an ability to recognize human naming conventions [21]. Animals including dogs (*Canis familiaris*) and chimpanzees (*Pan troglodytes*) can respond to (but not produce) acoustic labels for individuals and for hundreds of unique objects, with research identifying some physiological bases

of name recognition [22,23]. Such studies suggest that animals of diverse species can identify individuals (or objects) by name, much the same way that humans do.

Although many animals respond to human language, only a small number of species can learn to produce language-like sounds or use those sounds appropriately [24,25]. In particular, parrots are excellent at learning vocalizations, including human words, and can correctly apply words as labels [26–30]. This capacity for vocal production learning allows researchers to examine whether and how animals use vocal labels, rather than just respond to them, which provides a richer picture of the cognitive aspects of word use and labeling. Focused studies of grey parrots have shown that they incorporate their own names and the names of people and animals into their vocal repertoires, sometimes combining names with other words into multiple phrases [28,31]. Grey parrots recognize that parts of speech are distinct sounds that can refer to different concepts and can be recombined to create new meanings [32,33]. Together, these data suggest that grey parrots have the capacity to understand and appropriately use names. It is, however, unclear whether grey parrots are unusual in their ability to vocally label individuals using human naming conventions.

Research on the topic of name use by animals is generally limited to single-species studies with relatively small sample sizes, and many authors hesitate to conclude that documented vocal labels are names [8,28,31,33–35]. Large data sets provide the opportunity to assess potential name use by members of multiple species in many contexts. Further, studying name use within the framework of human language offers an accessible method for determining whether animals learn and use names as acoustic labels for individuals, as humans do [2]. Here we present a large-scale study describing how often parrots that live in companionship with humans learn names and when they use names around people and other animals. We assessed name use by parrots based on a survey of over one thousand companion birds, with relevant data reported for almost 900. We examined rates and contexts of name use among companion parrots of multiple species, and we looked for evidence that parrots correctly applied names as individual vocal labels. Our survey approach allowed for a broad, multi-species assessment of vocal labeling with clearly identifiable names.

## Materials and methods

Data were collected between October 5, 2020 and August 1, 2024 via the "What Does Polly Say? survey which asked people to report on their companion parrots ([21], https://www.manyparrots.org/). All research was conducted in accordance with University of Northern Colorado Institutional Review board policies with informed consent. Survey-takers voluntarily provided data via an online text-entry form and were informed that they could stop or withdraw at any time before submitting their responses. As this research involved only survey responses from humans about parrots, it did not require IACUC or IRB approval.

During our sampling period we collected reports on 1202 parrots of 89 total species. The survey asked for the names of the survey-taker and parrot, and included the following prompts: 1) "*Optional* Please list some of the words or phrases your parrot uses and their contexts" and 2) "Is there anything else you would like to tell us?" Some participants provided freeform answers to these prompts, listing words and phrases used by companion parrots of 78 species (S1 Table), often including names. When survey-takers answered the prompts above, we extracted all phrases that included recognizable human-created names or nicknames from the dataset. Included names a priori matched the general linguistic criteria of being proper nouns that multiple individuals would use to label a single individual. Our study included names used for people and for animals; we noted and excluded proper names that referred to objects or brands (i.e., Amazon Alexa, pet food brand names).

We examined what types of individual names birds learned and whether they used them in appropriate human contexts. To do that, we scored all parrot phrases that contained names for the name referent type and phrase use context as detailed in Tables 1 and 2. To create the name type and context categories we initially surveyed that data set for obvious category groupings. This led us to classify name types into four categories: the name of the vocalizing bird, name of a person, name of a companion bird, and the name of a non-bird pet (Table 1). We also included an unknown category when

**Table 1. Categories of name types used by parrots in spoken phrases.**

| Name type | Definition |
|---|---|
| Name of bird | The bird's own name or a variant of it, including nicknames, as indicated by the respondent or based on a match with the parrot's name |
| Name of person | The name or nickname of a human, as indicated by the respondent or based on a match with the name of the survey-taker |
| Name of companion bird | The name or nickname of another bird, as indicated by the respondent |
| Name of pet | The name or nickname of a pet other than a companion bird, as indicated by the respondent |
| Unknown name | A recognizable name but with no referent described |

**Table 2. Context categories for parrot phrases that contained names.**

| Specific Context | Definition |
|---|---|
| Greeting | The bird is reported to say this phrase containing a name when a human or animal arrives or is seen by the bird |
| Separation | The bird is reported to say this phrase containing a name when someone departs, is absent, or the bird goes to bed |
| Attention | The bird is reported to say this phrase containing a name to get attention from a human or animal |
| Request | The bird is reported to say this phrase containing a name when it wants something besides attention |
| Social | The bird is reported to say this phrase containing a name in social contexts that do not match the more specific categories above. This category includes talking to other pets (example: "when telling the dog to be quiet"), answering human prompts or questions (example: "when asked 'what's your name'"), and in other social interactions |
| Personal | The bird is reported to say this phrase containing a name when alone or not in an obviously social context (examples: "when it's raining" or "when he is angry") |
| **General Context** | |
| Vocal Label | The bird is reported to use names as individual vocal labels, but specific context is not indicated (example: "Can call everyone in the house by first name") |

no referent was provided (Table 1). When birds used multiple names or nicknames for the same individual, they were all scored as being names for that referent.

We classified phrase use context into six specific categories (greeting, separation, attention, request, social, and personal) that described when or how the bird was reported to use each phrase containing a name (Table 2). Context was not assigned based on the semantic content of the phrase, but rather on the contextual use of that phrase reported by survey-takers; generally, this was reported as part of a "when" phrase. For example, a bird [686] that says "goodnight [name]" "*when I leave her room for the night*" was scored in the separation context, but a different bird [56] that says

"goodnight [name]" "*when meeting people*" was scored in the greeting context (S2 Table). In some cases, survey respondents stated that the bird used a name correctly as a vocal label without describing the specific context. In those cases, we accepted that name was used as a "vocal label" and coded it as such (Table 2).

After creating the type and context categories and extracting all instances of name use, an observer scored the phrases containing names extracted from the data set for name type and context, following the definitions in Tables 1 and 2. A second observer checked those scores to verify the categorizations and a third observer resolved the small number of discrepancies. These observers then reviewed the data using a two-tiered system to examine whether the dataset contained evidence of parrots using names as appropriate and individualized vocal labels (Fig 1). We ensured that multiple people scored all phrases and came to consensus about each name type and context to improve coding reliability.

A parrot was scored as using a name "appropriately" if it was reported to use the name or nickname to correctly refer to a person or animal (i.e., saying "quiet Rufus" to a dog named Rufus when it barks). We note that this type of name usage does not necessarily imply that the parrot understands all parts of the phrase; it only recognizes that the bird uses the name phrase consistently in contexts that address or refer to the individual with that name. All instances of appropriate name use were additionally scored for "individualized" name use. A parrot was considered to use a name in an individualized fashion when survey-takers indicated that the parrot applied a proper name or nickname appropriately only to a single individual despite interaction with multiple individuals of that type (i.e., saying "quiet [name]" to different dogs when they bark and using the correct names for each). In these contexts, the use of that name was clearly not applied to a class of individuals or simply as part of a non-personalized phrase (e.g., "quiet Rufus" is not used generally to tell others to "be quiet"). Most examples of "individualized" name use came when a parrot substituted varied names into a longer phrase as appropriate, responded to or labeled multiple individuals, or when they asked for someone who was not present (suggesting an awareness of that individual's absence). We scored examples of "appropriate" and "individualized" name use to provide baseline evidence that parrots can use names to correctly communicate with or about individual humans and animals.

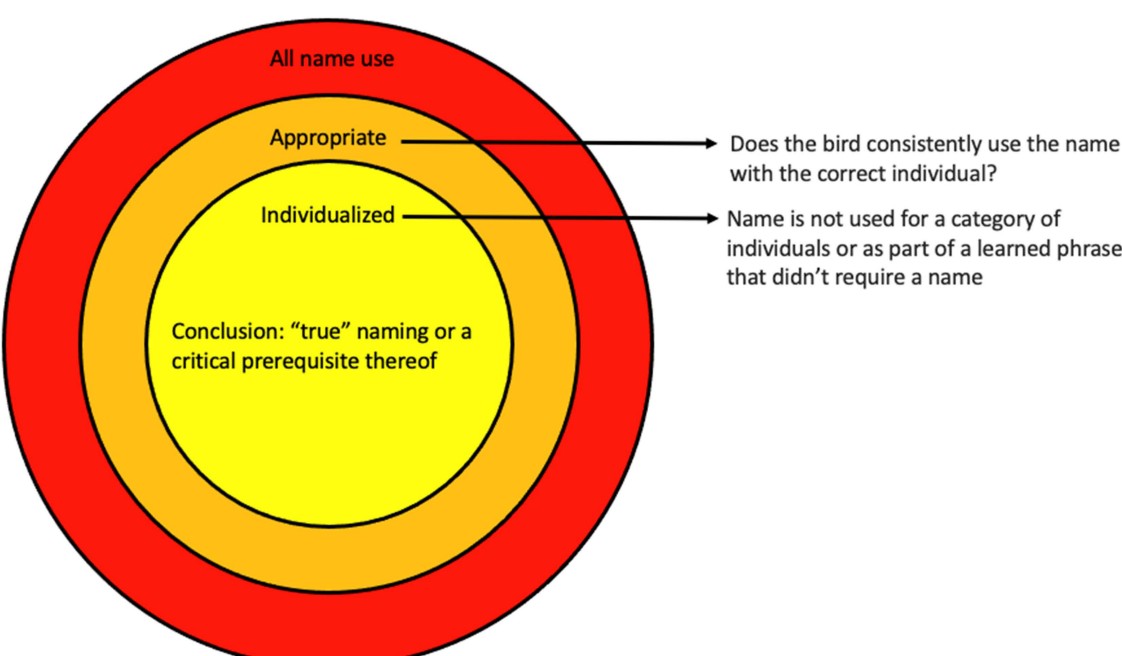

**Fig 1. Scoring scheme for appropriate and individualized name use by parrots, as reported by their human companions.**

A phrase was coded as appropriate name use when two of three scorers rated it as such. Two people then scored all phrases with appropriate name use for individualized name use and came to a consensus about whether each one represented individualized use of a proper name. Because survey-takers provided varying amounts of information about each bird and each vocalization (S2 Table), scoring for appropriate and individualized name use by parrots was not always clear cut. By having multiple coders and seeking agreement between them based on the above definitions, we ensured that our scoring was as standardized as possible, but we acknowledge that all scores are subjective. We further point out that the dataset is not perfectly representative of parrot name use, as survey respondents are more likely to report on vocalizations with meaning to them. Additionally, context information provided by the survey respondents may include biases in human interpretation of each parrot vocalization. Therefore, reports of correct, individualized name use do not unequivocally demonstrate that parrots understand names as vocal labels, but they offer a first step towards assessing this ability across a wide range of parrots. The scope of our survey-based sampling allowed for data collection on many more species than have been examined using more tightly controlled research methods.

We compiled descriptive statistics and rates of name use by parrots in varied contexts. We report those below along with examples of name use by parrots.

## Results

Survey participants reported on 1202 parrots and provided words and phrases used by 884 different birds, with names included in the examples for 413 of these (S2 Table). Thus, name examples came from 47% of all birds for which we had word and phrase use data. This reporting rate indicates widespread use of names by parrots that mimic human language, especially given that 47% is a minimum estimate because survey respondents did not provide full repertoires for each bird. Parrots reported to say names came from 63 different species, 81% of the 78 species for which words or phrases were reported (S1 Table).

Among the 413 parrots that used names, survey takers reported 801 example phrases that included human or animal names. Individual parrots of many species were reported to use multiple different phrases that included names for people and animals (Table 3). Some of these were names as stand-alone phrases, and others included a name in addition to other words (S2 Table). Ninety-six parrots from 40 species used more than one phrase that included the same name paired with other human words (i.e., "hi Polly" and "bedtime for Polly"). One hundred and forty-five parrots of 44 species were reported to use more than one unique name, often their own name plus the name(s) of humans and other pets in the household. Many parrots also used multiple variants of a name or nicknames, similarly to human use of nicknames. We were unable to conclusively identify all nicknames from the data provided, but a conservative estimate is that at least 26 parrots from 19 species used nicknames, as we saw obvious name modifications in different phrases from those individuals (i.e., [618] "Quince!" and "The Quincenator"). Twenty-six parrots of fifteen species used four or more different names (including nicknames) in their phrases with one knowing at least nine individual names.

**Table 3. Number of birds reported to use 1-6+ unique phrases containing names (Range 1-12 phrases).**

| # Name phrases | # birds | # species |
|---|---|---|
| 1 | 223 | 50 |
| 2 | 91 | 32 |
| 3 | 50 | 26 |
| 4 | 31 | 18 |
| 5 | 9 | 8 |
| 6+ | 10 | 5 |

Most name phrases used by parrots included their own names, but many birds were also regularly reported to use the names of humans, other birds, and non-avian pets (primarily dogs; Table 4). Some parrots were reported to use proper names for objects such as Amazon's Alexa (4 birds), individual toys (1 bird), or food brand names (1 bird), but we coded those as object labels and excluded them from summary analyses. Interestingly, no parrots were reported to use proper names for places (e.g., Austria), a type of proper name that humans use frequently [2].

Survey respondents provided specific context information for 230 reported name phrases and indicated that another 43 were used as vocal labels (see Table 2 for definitions) (Table 5). Parrots were reported to use phrases containing names in a range of social situations, including when seeking attention or objects, greeting others, or separating from others (Table 5).

Parrots used all name types in most of our contextual categories (Fig 2). In greeting, separation, attention, and request contexts, parrots most often used their own names and the names of people (Fig 2). Other types of social interactions prompted a range of name type use, as did vocal labeling (Fig 2). Parrots were regularly reported to use names in incorrect human contexts (Fig 2). For example, many birds said their own names when seeking attention or in greetings (Fig 2). Often these phrases appeared to be mimicry of phrases that humans use to get the attention of or greet the bird. For example, many birds said "Hi [bird's own name]" when greeting someone. Other birds used names as vocal labels for classes of individuals rather than single individuals; for example, using the name of one dog to refer to another dog or all dogs. In some cases, survey-takers reported directed communication towards a receiver that didn't involve names, such as using a "flock whistle" with certain communication partners. Similarly, a bird might say the same phrase (its own name)

Table 4. Name type category assignments indicating the number of reported phrases that contained each name type and the number of birds and species that used each name type.

| Name Type | # phrases | # birds | # species |
|---|---|---|---|
| Name of bird | 413 | 280 | 55 |
| Name of person | 166 | 124 | 22 |
| Name of companion bird | 66 | 43 | 36 |
| Name of pet | 56 | 39 | 14 |
| Unknown name | 100 | 72 | 25 |
| Total | 801 | 413* | |

*413 parrots were reported to use names – this column sums to 558 because many birds used names of multiple types.

Table 5. Context use assignments for all phrases, birds, and species for which name use context was provided by survey-takers.

| Context | # phrases | # birds | # species |
|---|---|---|---|
| Greeting | 38 | 32 | 17 |
| Separation | 39 | 29 | 12 |
| Attention | 34 | 32 | 19 |
| Request | 21 | 17 | 11 |
| Social | 80 | 57 | 29 |
| Personal | 18 | 15 | 11 |
| Vocal label | 43 | 29 | 14 |
| Total | 273 | 170* | |

*Name use contexts were reported for 170 parrots – this column sums to 211 because some birds used names in multiple contexts

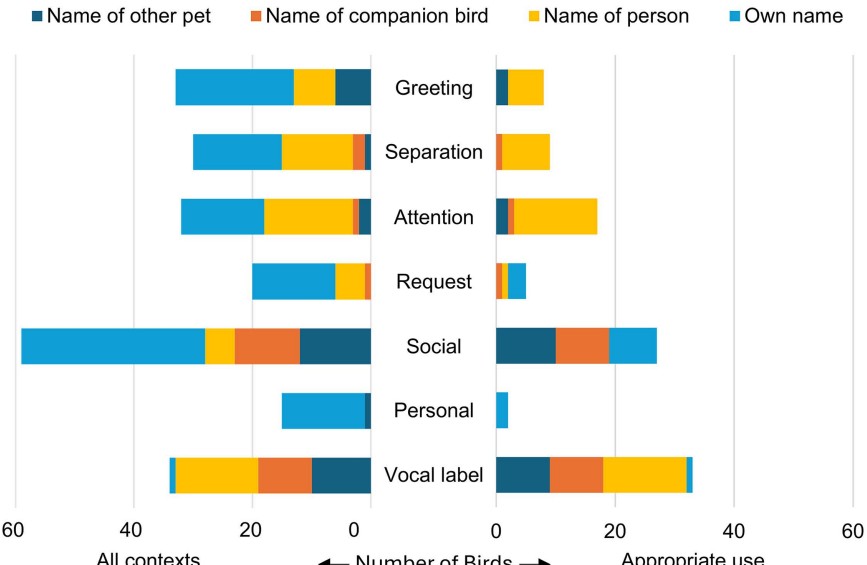

**Fig 2. Number of parrots that used each name type in each contextual category for all reported name vocalizations with context (left) and for appropriate name use (right).**

in a voice that mimicked different humans when it interacted with each of them. These examples represent individualized communication but not name use as defined in this study.

Although many names were used by parrots incorrectly or ambiguously, others were reported to be applied correctly. We identified 131 examples of appropriate name use from 88 different parrots of 30 species. This represents 48% of the phrases for which context was scored, 52% of individuals with name phrase contexts reported, and 68% of species with name phrase contexts reported. Most appropriate name use came when parrots referred to other individuals, including humans and animals (Fig 2). This stands in contrast to the total set of name phrases, which predominantly included the parrots' own names (Fig 2). When using names appropriately to seek attention or greet and separate from others, parrots most often used the names of people (Fig 2). Examples included parrots using names for humans, other birds, non-bird animals, and themselves appropriately in a variety of social interactions (Fig 2, S2 Table).

Among the appropriately used name phrases, we coded 69 examples of individualized name use from 42 parrots of 19 species. For some of these examples, survey takers indicated something general like "Polly uses the names of all house-hold members correctly" (context coded as "vocal label") while others provided specific context that indicated individualized name use. For example, one parrot said "quiet [parrot name]" to multiple other parrots in its flock as appropriate when each made noise (context coded as "social"). Others called multiple individuals (often dogs and/or humans) by name when they saw them (context coded as "greeting") or wanted attention (context coded as "attention"). Several birds were reported to learn the names of visitors (humans and birds) and use them correctly. At least ten birds were reported to ask for specific people by name only when they were not present (context coded as "separation"), suggesting individual and situational awareness in name use plus a concept of individual permanence. One parrot that used its own name in social interactions was reported to correct people who called it the wrong name by telling them its name.

Most species in our dataset were represented by fewer than 10 birds for which we had name use context information (S2 Table). For five species, however, we sampled at least ten birds that used names in stated contexts. Table 6 shows the percentage of individuals of each species for which context descriptions indicated appropriate and individualized name use. Grey parrots (*Psittacus erithacus*) had higher rates of appropriate and individualized name use than other species.

**Table 6. Rates of appropriate and individualized name use among species in which contexts were reported for at least ten birds.**

| Species | n | % with appropriate name use | % with individualized name use |
|---|---|---|---|
| *Amazona aestiva* | 10 | 30% | 0% |
| *Amazona oratrix* | 11 | 45% | 18% |
| *Myiopsitta monachus* | 12 | 42% | 17% |
| *Psittacus erithacus* | 45 | 67% | 38% |
| *Pyrrhura molinae* | 14 | 36% | 7% |

## Discussion

Our survey of human vocal mimicry by parrots revealed widespread learning of names. Parrots can learn to pronounce the names of multiple social companions in human languages (primarily English for this data set). This allowed us to assess how parrots of many species use names, revealing that they learn to say their own names as well as the names of humans and animals. Many parrots in the survey clearly learned to mimic names without understanding them as vocal labels (e.g., treating the name as part of a greeting), but many others used names in appropriate contexts and were reported to apply them correctly as individual identifiers. Results are concordant with other research on parrot word use and consistent with the idea that animals are capable of using names as vocal labels, as has been hypothesized to occur among wild animals [31,36].

Names are ubiquitous in human social groups and each name is a distinctive referential set of sounds that all group members apply to one individual [1,33]. Children learn to distinguish proper names from object category labels early in life, reinforcing the conceptual and cognitive distinctiveness of names [37–39]. In this study, we identified names vocalized by hundreds of parrots of 63 different species. The frequency with which names appeared in our dataset suggests that they are in some way salient to parrots who live with humans. We were unable to control for the social and learning environments of our subjects, making it difficult to know why so many birds learned names, but we expect that names represent an important part of human-parrot communication. Our results fit with a previous study which found that names represent more than a quarter of the units in grey parrot vocabularies, a rate similar to what is seen in young children [31]. When considering that parrots are likely exposed to much less language teaching and enrichment than children are [40], this highlights the importance of names, and thus individual recognition and labeling in human-parrot vocal interactions.

Parrots in our study population most often said their own names and regularly used the names of people and animals, but never learned other types of proper names, such as place names. This likely reflects name use patterns by the humans who interact with parrots. Humans undoubtedly say their parrots' names to those individuals frequently, providing extensive opportunity and targeted communication within which parrots learn their names. The names of humans and other pets are less likely to be said to parrots directly, and many of the name phrases used by parrots implied that they learned them from listening to humans talking to each other or to animals. For example, a parrot that says "quiet [name of dog]" when the dog barks most likely learned from a human who was directing their speech at the dog. Thus, our results suggest that parrots learn names both when humans are directly interacting with them and by eavesdropping on other communication pairings. This matches observations showing that grey parrots can learn language directly from a trainer or from a model interacting with the trainer, where the bird was an observer [41]. Some of the name phrases in our dataset were likely not intentionally taught (for example, "Quiet [dog name]" or "Alexa order water"), highlighting the ability of parrots to independently learn names as parts of speech, and to learn by eavesdropping on communication events that don't target them as a receiver. This skill is expected to be valuable in natural settings where individuals integrating into new social flocks must learn local dialects and contact calls by listening to established flock members as they signal to each other [42,43].

Our data revealed that parrots use names in varied ways, most of them relating to social interactions, whether that be greetings and separations or responding to social cues and vocal prompts. Very few parrots used phrases that included names to refer to the world at large, for example by learning place names. We also collected limited examples of parrots using names in non-social "personal" contexts, such as saying their own name when they make a particular movement. This pattern matches the expectation that individual vocal labels are an important mechanism of social communication [8,16]. At the same time, this pattern might reflect biases in reporting by humans who tend to report examples of name use that are most salient in human interaction contexts.

Many parrots in our sample learned to say the names of several other individuals and even used nicknames, such that they might know multiple names for one individual. Further research in controlled settings should investigate whether parrots can appropriately use multiple vocal labels for one individual, as nicknames would suggest. Names were sometimes used by parrots in our data set as standalone phrases and sometimes as parts of longer phrases. Thus, further research should also investigate if and when parrots recognize the name subunits of their spoken phrases as separate from the phrases as a whole. Captive grey parrots can understand naming conventions and segment phrases into parts [28,32]. Segmenting phrases into parts has rarely been examined among other non-human animals, but may be important in natural parrot vocalizations, as consonant- and vowel-like subcomponents have been observed within individual budgerigar song notes, and wild yellow-naped amazons follow syntactical rules during duet interactions [44–46].

All of the names and name phrases we documented among parrots might be merely mimicked with no cognitive understanding of what a name is or, alternatively, might represent true vocal labeling. By matching contextual name use with phrase content we sought evidence that names represent specific individuals in the minds of the parrots. For most name phrases in our data set we lacked context information and were therefore unable to determine if parrots used them appropriately. For other phrases we had evidence that parrots did not use names as correct referential labels; only about half of name phrases with context provided were reported to be used appropriately. Thus, parrots often learn to mimic the sounds of names without understanding their typical human meanings, and survey-takers regularly reported that. We posit that some of these "mistakes" might represent situations where parrots learned to use names in ways that are functional, even if not semantically correct or reflective of human learning patterns. For example, parrots used their own names in many phrases with an "attention" context. This result is similar to that of a previous study which found that the names most commonly used by grey parrots are their own [31]. Although a parrot that says its own name when it wants attention is not using the name in a typical adult human linguistic context, the name still provides a benefit to the bird if its human responds with attention. In this way, much of the "inappropriate" name use reported by survey takers could be functional to the parrot in promoting desired social interactions, reflecting the substantial vocal and social learning abilities of parrots [47,48]. In addition, this type of third-person self-reference (illeism) is characteristic of very young children, which often go through a period of referring to themselves by their own name, possibly due to the difficulty of mastering pronouns [49]. Parents will also naturally use third-person speech when communicating with young children to simplify language and aid learning [50]. The possibility that parrots may be using names in a manner akin to that of young children warrants further research. Interestingly, this type of self-labeling also resembles the behavior of wild parrots and dolphins, in which individually distinctive, self-identifying vocal signatures have been documented in several species and are most often produced by the referent [18,51–53].

While not all companion parrots in our data set were reported to use proper names following human language conventions, many did (about half of all individuals for which we had contextual use information). Parrots of 30 species were reported to use names appropriately, with grey parrots showing the strongest tendencies towards correct name use. Thus, our data offer preliminary evidence that parrots of diverse species may understand the concept of a name as it is used by humans, and that species differences may exist in learning and cognition as it relates to individual labeling [3,26,41]. It also lends support to the previously demonstrated ideas that parrots can recognize the individuality of others and are

conscious of themselves as unique beings [54–56]. In the cases we documented, the names used were vocal labels widely recognizable to humans, and many of the birds indicated individualized name use by substituting multiple names into a phrase following human linguistic conventions. For example, one bird said "goodnight [name]" to each flock mate as they were put to bed, regardless of order. In this case, each name phrase clearly refers to a single bird. Although our survey-based method prevents us from definitively establishing that a bird using those phrases understands the concepts of "goodnight" and "individual label", the patterns are consistent with that interpretation. Further, such behavior matches the understanding that wild parrots also communicate about roosting behavior prior to sleeping [27] and use vocalizations to address individuals or flock-mates [16,57,42]. Most wild parrots live in complex fission-fusion societies and are expected to benefit from such personalized communication [48]. Thus, traits related to learning ability and social integration likely underlie the name-use capabilities we documented, and they provide clear evolutionary potential for the use of names in nature.

Researchers studying a variety of animal taxa have identified features of their vocalizations that reflect the identity of the signaler and receiver, but clear evidence for animal use of names (consistent vocal labels for non-self individuals) has been elusive [6,8,19]. In several species, researchers note that vocalization tone, frequency, amplitude, and other acoustic cues may be modulated to get the attention of certain receivers [8,10,15]. We observed a potential analog to that in our data set, which included examples of parrots that mimicked the voices of different individuals when speaking to them. Voice or other vocal modulation can be used to target individual signal receivers without the use of a name in the human sense [58,59]. Studies examining vocal identification via call modulation offer exciting results, but the name-like features of those calls are more difficult to interpret than stand-alone human names used by parrots. In English, human names do not depend on tone or voice modulation; they are consistent and identifiable sequences of phonemes most often used by non-self-individuals. Here we demonstrated those features clearly; parrots in our study learned human-given names frequently and used them to appropriately refer to other individuals.

## Conclusion

Scientists and the public alike have often wondered whether animals can use proper names for themselves and each other [60]. Compelling evidence indicates that many animals can recognize and respond to human-given names [22,23], while others can invent and use individual vocal signatures [8,10,16,18]. No previous study, however, has provided evidence that a speciose group of animal subjects can produce and appropriately use names that are recognized as such according to human linguistic conventions [3]. We took advantage of the fact that parrots can mimic human speech to document evidence of name use by multiple avian species. Results indicated that parrots often learned names from their humans and used them in a variety of contexts, some of which are consistent with the ability to cognitively associate a name with an individual. All birds in our data set used human-derived names, leaving open questions about whether and how they might create names themselves. Results clearly demonstrate, however, that animals can learn and use proper names in appropriate social situations. Further work should be done in controlled settings to better understand the cognitive underpinnings of this behavior in parrots and beyond. We expect that the ability to label individuals is present in wild animals as well as captive ones. We hope that future work will find accurate ways to identify animal naming via methods that do not depend on human language.

## Supporting information

**S1 Table. Species name use.** Parrot species reported to use names, including lists for all species with words and phrases reported, species that mimicked names, species that showed appropriate name use, and species that showed individualized name use.
(XLSX)

**S2 Table. Dataset of name phrases with coding.** Dataset of parrot name phrases included in this study, with de-identified supporting information.

(XLSX)

## Acknowledgments

We thank Alexandra Charles, Tahais Guerrerro-Rocha, Amira Brockington, and Sandis Walter for organizing and scoring data. We received very helpful manuscript reviews from Dr. Nicolas Giret and three anonymous reviewers. Most importantly, we thank all the survey-takers who provided information about their parrots.

## Author contributions

**Conceptualization:** Lauryn Benedict, Viktoria Groiss, Marisa Hoeschele, Eva Reinisch, Christine R Dahlin.

**Data curation:** Lauryn Benedict, Viktoria Groiss.

**Formal analysis:** Lauryn Benedict, Viktoria Groiss.

**Funding acquisition:** Marisa Hoeschele.

**Methodology:** Lauryn Benedict, Viktoria Groiss, Marisa Hoeschele, Eva Reinisch, Christine R Dahlin.

**Visualization:** Lauryn Benedict, Marisa Hoeschele.

**Writing – original draft:** Lauryn Benedict.

**Writing – review & editing:** Lauryn Benedict, Viktoria Groiss, Marisa Hoeschele, Eva Reinisch, Christine R Dahlin.

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
