## [Decision Letter · Decision Letter 0]

11 Jan 2026

Dear Dr. Benedict,

Thank you for submitting your manuscript to PLOS ONE. After careful consideration, we feel that it has merit but does not fully meet PLOS ONE’s publication criteria as it currently stands. Therefore, we invite you to submit a revised version of the manuscript that addresses the points raised during the review process.

We look forward to receiving your revised manuscript.

Kind regards,

Jianhong Zhou

Staff Editor

PLOS One

Journal Requirements:

“This work was supported by the Vienna Science and Technology Fund (WWTF) project ANIML (LS23-014) to MH”

Additional Editor Comments:

Please note that we have only been able to secure a single reviewer to assess your manuscript. We are issuing a decision on your manuscript at this point to prevent further delays in the evaluation of your manuscript. Please be aware that the editor who handles your revised manuscript might find it necessary to invite additional reviewers to assess this work once the revised manuscript is submitted. However, we will aim to proceed on the basis of this single review if possible.

Reviewers' comments:

Reviewer's Responses to Questions

**Comments to the Author**

1. Is the manuscript technically sound, and do the data support the conclusions?

Reviewer #1: Partly

2. Has the statistical analysis been performed appropriately and rigorously?

Reviewer #1: N/A

3. Have the authors made all data underlying the findings in their manuscript fully available?

Reviewer #1: Yes

4. Is the manuscript presented in an intelligible fashion and written in standard English?

Reviewer #1: Yes

Reviewer #1: In this study, the authors investigate the ability of companion parrots to use individual names. They rely on an online survey with the manyparrots.org platform. The dataset includes 1202 birds representing 64 parrot species. The authors proposed distinct categories of name types and contexts in which the names were uttered and tried to assess whether the names were used appropriately. They report that 413 parrots from 30 distinct species used names and that 88 different parrots used these names appropriately. The authors addressed several points in the discussion related to the learning ability, to the putative distinction between using name alone or in sentences, to whether parrots understood the meanings of using the name and on the particular acoustic features that may underline these names.

While I found the paper original and interesting, I have several concerns that should be addressed by the authors. First, in the introduction, I found that details are missing in the paragraph on individual signature, especially regarding all the literature in songbirds (some papers: Elie JE, Theunissen FE (2018) Zebra finches identify individuals using vocal signatures unique to each call type. Nat Commun 9:4026; Lehongre K, Aubin T, Robin S, Negro CD (2008) Individual Signature in Canary Songs: Contribution of Multiple Levels of Song Structure. Ethology 114:425–435; Gidl H, Binder S, Osiecka AN, Klump BC. The ontogeny of vocal identity in carrion crows (Corvus corone). Anim Cogn. 2025 Dec 16. doi: 10.1007/s10071-025-02021-5. Epub ahead of print. PMID: 41402471; Diniz P, Silva-Jr EF, Rech GS, Ribeiro PHL, Guaraldo AC, Macedo RH, Amorim PS. Duets convey information about pair and individual identities in a Neotropical bird. Curr Zool. 2024 Oct 21;71(4):456-468. doi: 10.1093/cz/zoae064. PMID: 40860766; PMCID: PMC12376049).

I am also a bit sceptical on the power of the results since it is mostly based on anecdotic examples. I value the work done by the authors to honestly quantify (and report) the use of names by the parrots, but whether this name use is very peculiar in the repertoire of these birds remains unclear to me compared to any other words the parrots are able to imitate. Relying on an online survey is clearly important for this kind of study, but I am at the same time wondering about the bias in the responses provided by the respondents. Especially, I would not be so surprised that the respondents were more willing to report examples when their own parrot(s) used names appropriately, than incorrectly. This putative bias should be discussed to temper the results. It is also a bit frustrating that no discussion is made at the species level: the authors go from examples at the level of single individuals to “parrots” in general, without mentioning whether name use was more frequent in any specific species (I do understand that there is a limitation due to the number of individuals per species, but this could be discussed a bit).

Below are more specific comments, not sorted by importance:

Materials and Methods

Please provide the number of species and the corresponding number of birds per species.

From table S2, I counted 63 different species reported to use words, not 64 as indicated in the main text and in table S1: after a check, Poicephalus gulielmi is reported twice in table S1.

In table S2, the species name was not provided for a few birds (bird IDs: 119, 553). And no genus, nor species were provided for a few other birds (birds IDs:485, 576, 3136) so I am wondering why these birds were included in the dataset.

Table 2: it is not very clear to me how the authors were able to properly separate the different contexts. For example, between “greeting” and “attention”.

L. 184: Please provide the corresponding 88 species in Table S1

L. 187: how many parrots used nine different phrases?

L189, 191 and 194: please provide the corresponding number of species of the 91 parrots using multiples phrases that include names, of the 145 parrots using more than one unique name and of the 21 parrots using 4 or more different names. This could be also provided in the Table 4.

L190: please provide a real number of parrots using multiple variants of names, and how many variants

L202/203: “Interestingly, no parrots were reported to use proper names for places (e.g. Austria), a type of proper name that humans use frequently (2)”. This should be either further discussed or removed. I do not see why it is interesting: my guess would be that animal caretaker do not mention places names very often when interacting with their bird (or do you have any assessment to argue for it?).

L. 221 : “Parrots were regularly reported to use names in incorrect human contexts. For example, many birds said their own names when seeking attention or in greetings.” Please provide numbers. Are these errors done more frequently by some specific species?

Figures 2 and 3: please add a label on the X axis (number of parrots). For clarity and allow direct comparison, these two figures could be combined in a single figure using a mirrored histogram.

§ starting at L. 235: it would be interesting, if possible, to get a score of how often individual parrots used a name appropriately vs non-appropriately: it is not the same if an individual uses ten times an appropriate name and ten times an inappropriate name (score of appropriateness= 50%) than a bird using four times an appropriate name and one times an inappropriate name (score of appropriateness= 80%).

.

Reviewer #1: **Yes:** Nicolas GiretNicolas GiretNicolas GiretNicolas Giret

---

## [Author Response · Author response to Decision Letter 1]

23 Feb 2026

Dear PLOS ONE Editorial Team and Dr. Giret,

Thank you for your very helpful comments on our manuscript. We catalog our responses to those comments below. All line numbers refer to the tracked-changes document,

As a general overview, we have revised the results reporting to provide more species-specific information in the text and tables. We have also updated some numbers and made an effort to recognize both the benefits and limitations of survey data like ours. Finally, we added several references (changes not tracked in the references section because the Zotero plug-in doesn’t track them), reviewed manuscript formatting and made edits to meet PLOS ONE style requirements.

Reviewer #1 Comments: In this study, the authors investigate the ability of companion parrots to use individual names. They rely on an online survey with the manyparrots.org platform. The dataset includes 1202 birds representing 64 parrot species. The authors proposed distinct categories of name types and contexts in which the names were uttered and tried to assess whether the names were used appropriately. They report that 413 parrots from 30 distinct species used names and that 88 different parrots used these names appropriately. The authors addressed several points in the discussion related to the learning ability, to the putative distinction between using name alone or in sentences, to whether parrots understood the meanings of using the name and on the particular acoustic features that may underline these names.

While I found the paper original and interesting, I have several concerns that should be addressed by the authors. First, in the introduction, I found that details are missing in the paragraph on individual signature, especially regarding all the literature in songbirds (some papers:

- Elie JE, Theunissen FE (2018) Zebra finches identify individuals using vocal signatures unique to each call type. Nat Commun 9:4026

- Lehongre K, Aubin T, Robin S, Negro CD (2008) Individual Signature in Canary Songs: Contribution of Multiple Levels of Song Structure. Ethology 114:425–435

- Gidl H, Binder S, Osiecka AN, Klump BC. The ontogeny of vocal identity in carrion crows (Corvus corone). Anim Cogn. 2025 Dec 16. doi: 10.1007/s10071-025-02021-5. Epub ahead of print. PMID: 41402471

- Diniz P, Silva-Jr EF, Rech GS, Ribeiro PHL, Guaraldo AC, Macedo RH, Amorim PS. Duets convey information about pair and individual identities in a Neotropical bird. Curr Zool. 2024 Oct 21;71(4):456-468. doi: 10.1093/cz/zoae064. PMID: 40860766; PMCID: PMC12376049).

Response: Thank you for this point. There is, indeed, a very large and compelling literature on individual recognition via acoustic cues among birds. The references you provided exemplify that, as does the long literature on neighbor-stranger discrimination by wild birds. Following your suggestion, we now make this clear in lines 58-61 where we have added the four citations that you suggested. We do not discuss this topic in depth, however, because individual acoustic signatures are not truly name-like; they allow a listener to identify a vocalizing individual, but they don’t allow a vocalizing individual to identify or target a listener, as names do. We now emphasize that distinction and our rationale for focusing our text around the literature on bats, elephants, dolphins, and parrots that shows how animals acoustically identify non-self individuals with their vocalizations.

I am also a bit sceptical on the power of the results since it is mostly based on anecdotic examples. I value the work done by the authors to honestly quantify (and report) the use of names by the parrots, but whether this name use is very peculiar in the repertoire of these birds remains unclear to me compared to any other words the parrots are able to imitate. Relying on an online survey is clearly important for this kind of study, but I am at the same time wondering about the bias in the responses provided by the respondents. Especially, I would not be so surprised that the respondents were more willing to report examples when their own parrot(s) used names appropriately, than incorrectly. This putative bias should be discussed to temper the results.

Response: We share these concerns and acknowledge that survey data is not entirely reliable. At the same time, it is the only way to assess behavior across so many individual birds, and we think that benefit makes it worth conducting this research, as long as the caveats you point out are reported. We do so in several places in the manuscript as follows:

Lines 183-186 (Methods) we “point out that the dataset is not perfectly representative of parrot name use, as survey respondents are more likely to report on vocalizations with meaning to them. Additionally, context information provided by the survey respondents may include biases in human interpretation of each parrot vocalization.”

Lines 339-340 (Discussion) we added a sentence indicating that results “might reflect biases in reporting by humans who tend to report examples of name use that are most salient in human interaction contexts.”

We also point out that we recorded many instances of parrots using names in incorrect human contexts (lines 356-358), confirming that survey-takers didn’t only report examples of correct name use. In our discussion of this topic and other results throughout the manuscript, we have chosen our words very carefully to indicate that our results are consistent with an understanding of the concept of names by parrots, but not ironclad evidence of that.

It is also a bit frustrating that no discussion is made at the species level: the authors go from examples at the level of single individuals to “parrots” in general, without mentioning whether name use was more frequent in any specific species (I do understand that there is a limitation due to the number of individuals per species, but this could be discussed a bit).

Response: You are correct that we avoided most discussion of this topic because our sampling was limited for most species. To better indicate that and to address questions about differences between species, we added species level sampling numbers to the following:

- Table 4 now indicates how many species used each name type (bird, person etc.)

- Table 5 now indicates how many birds and species used names in each context (greeting, separation, etc.)

- Table S2 now shows the number of individuals of each species that learned names (column c), used them appropriately (column f), and used them individually (column i)

We also looked at species for which we had reasonable sampling and could potentially calculate the tendency for individuals to use names appropriately. The following table (now table 6 in the manuscript) shows all species (only 5) for which we had at least 10 individuals that used names in known contexts. We had many species with more than 10 individuals that used names, but the context data set was limiting. Among those species, grey parrots were more often reported to use names correctly according to human conventions. We have added this to the manuscript in lines 285-293 and 379-382.

Context Appropriate Individualized

Amazona aestiva 10 30% 0%

Amazona oratrix 11 45% 18%

Myiopsitta monachus 12 42% 17%

Psittacus erithacus 45 67% 38%

Pyrrhura molinae 14 36% 7%

Below are more specific comments, not sorted by importance:

Materials and Methods

Please provide the number of species and the corresponding number of birds per species.

Response: We now list the number of birds and species included in the full sample on line 114. Lines 200-208 provide more details on sampling, and we have added the number of birds per species to table S1. We didn’t put that in the main text due to space constraints; it includes sample sizes for 63 different species.

From table S2, I counted 63 different species reported to use words, not 64 as indicated in the main text and in table S1: after a check, Poicephalus gulielmi is reported twice in table S1.

Response: Thank you for catching this. We have corrected the table and the text to say 63.

In table S2, the species name was not provided for a few birds (bird IDs: 119, 553). And no genus, nor species were provided for a few other birds (birds IDs:485, 576, 3136) so I am wondering why these birds were included in the dataset.

Response: These are birds for which the survey-takers did not provide species names. Each survey taker must have indicated in response to another question that they were reporting on a parrot that mimics human sounds. Therefore we included these parrots in the data set even if species wasn’t known.

Table 2: it is not very clear to me how the authors were able to properly separate the different contexts. For example, between “greeting” and “attention”.

Response: We have added text to clarify this in lines 139-142. Like with the other scoring, we had multiple reviewers look at these categories, and we emphasize that in lines 153-155. We acknowledge that these scores are subjective, but we attempted to minimize that as much as possible. At the same time, in many cases the scoring was unambiguous because survey-takers indicated that the bird used each phrase in contexts such as “when she wants my attention [attention]” or “when she sees me in the morning [greeting].”

L. 184: Please provide the corresponding 88 species in Table S1

Response: We have added this information to Table S1. We appreciate you asking us to list these because in doing so, we realized that we had a typo in the text and the total number of species with words or phrases reported in the survey was 78, not 88. This is now corrected in the text.

L. 187: how many parrots used nine different phrases?

Response: Just one. This is in a later sentence, line 214. We removed the mention of “nine phrases” in line 187 (now 203) to allow the later statement to stand on its own more clearly.

L189, 191 and 194: please provide the corresponding number of species of the 91 parrots using multiples phrases that include names, of the 145 parrots using more than one unique name and of the 21 parrots using 4 or more different names. This could be also provided in the Table 4.

Response: We have added this to the text in lines 205-207. In reviewing the numbers, we found an error: 96 (not 91) parrots used the same name in multiple phrases. We have corrected that and added species numbers for multiple name use. This is slightly different from what is in Table 4, as that indicates the number of total name phrases from each bird.

L190: please provide a real number of parrots using multiple variants of names, and how many variants

Response: We have added this information in lines 210-214. In calculating this we revised the text to indicate that we include cases where those 4 names included apparent nicknames. We now also report the species numbers for that in line 213.

L202/203: “Interestingly, no parrots were reported to use proper names for places (e.g. Austria), a type of proper name that humans use frequently (2)”. This should be either further discussed or removed. I do not see why it is interesting: my guess would be that animal caretaker do not mention places names very often when interacting with their bird (or do you have any assessment to argue for it?).

Response: We agree with you that the lack of place name learning likely reflects what owners say to their parrots. We think this is am important aspect of phrase learning and have added text to the discission point this out (lines 317-318).

L. 221 : “Parrots were regularly reported to use names in incorrect human contexts. For example, many birds said their own names when seeking attention or in greetings.” Please provide numbers. Are these errors done more frequently by some specific species?

Response: These numbers are in Figure 2, so we now refer to that at the end of this sentence (line 226). We did not have a way to accurately quantify “error rates” in name use, and instead are discussing general patterns here. Because of that and the fact that we have limited context information, we are unable to calculate meaningful “error rates” by species.

Figures 2 and 3: please add a label on the X axis (number of parrots). For clarity and allow direct comparison, these two figures could be combined in a single figure using a mirrored histogram.

Response: We have updated the figure as suggested, to create a new Figure 2.

§ starting at L. 235: it would be interesting, if possible, to get a score of how often individual parrots used a name appropriately vs non-appropriately: it is not the same if an individual uses ten times an appropriate name and ten times an inappropriate name (score of appropriateness= 50%) than a bird using four times an appropriate name and one times an inappropriate name (score of appropriateness= 80%).

Response: This would be very interesting to know! Unfortunately, we don’t have those data with our dataset. We could do follow-up research where we contact the survey participants to ask them this question. Thank you for that good suggestion.

Response: We have reformatted following these guidelines. Our figures are 6x4 inches, in .tiff format which should rescale cleanly.

Response: We have added information to our ethics statement and moved it to the methods section, lines 108-113.

Response: We have removed this statement from the manuscript.

“This work was supported by the Vienna Science and Technology Fund (WWTF) project ANIML (LS23-014) to MH”

Response: We have added the statement above to the cover letter.

5. Your ethics statement should only appear in the Methods section of your manuscript. If your ethics statement is written

---

## [Decision Letter · Decision Letter 1]

24 Mar 2026

Name use by companion parrots

PONE-D-25-58742R1

Dear Author,

We’re pleased to inform you that your manuscript has been judged scientifically suitable for publication and will be formally accepted for publication once it meets all outstanding technical requirements.

Kind regards,

Javed Iqbal, PhD

Academic Editor

PLOS One

Additional Editor Comments (optional):

Reviewers' comments:

Reviewer's Responses to Questions

**Comments to the Author**

Reviewer #1: All comments have been addressed

2. Is the manuscript technically sound, and do the data support the conclusions?

Reviewer #1: (No Response)

3. Has the statistical analysis been performed appropriately and rigorously?

Reviewer #1: Yes

4. Have the authors made all data underlying the findings in their manuscript fully available?

Reviewer #1: Yes

5. Is the manuscript presented in an intelligible fashion and written in standard English?

Reviewer #1: Yes

Reviewer #1: I would like to thank the authors for their revision. I just have minor comment:

I fell that the position of the sentence at line L. 58-59 ("Diverse species of birds can recognize individuals by the acoustic properties of their vocalizations (much as humans do when recognizing voices") is a little strange because it is mentionning ‘birds’ here while the subsequent phrases are about mammals (from L. 61). I would either move it or include some mammals references here and replace 'birds' by 'animals'.

.

Reviewer #1: **Yes:** Nicolas GiretNicolas GiretNicolas GiretNicolas Giret

---

## [Editor Report · Acceptance letter]

PONE-D-25-58742R1

PLOS One

Dear Dr. Benedict,

I'm pleased to inform you that your manuscript has been deemed suitable for publication in PLOS One. Congratulations! Your manuscript is now being handed over to our production team.

Kind regards,

on behalf of

Dr. Javed Iqbal

Academic Editor

PLOS One